# Baby HABIT-ILE intervention: study protocol of a randomised controlled trial in infants aged 6–18 months with unilateral cerebral palsy

Astrid Carton de Tournai [ID],[1] Enimie Herman [ID],[1] Estelle Gathy [ID],[1,2] Daniela Ebner-Karestinos [ID],[1,3] Rodrigo Araneda [ID],[1,3] Laurence Dricot [ID],[1,4] Benoît Macq [ID],[1,5] Yves Vandermeeren,[2,4] Yannick Bleyenheuft [ID] [1,4]

ACdT and EH are joint first authors.

For numbered affiliations see end of article.

**Correspondence to**
Dr Yannick Bleyenheuft;
Yannick.bleyenheuft@uclouvain.be

## ABSTRACT

**Introduction** Research using animal models suggests that intensive motor skill training in infants under 2 years old with cerebral palsy (CP) may significantly reduce, or even prevent, maladaptive neuroplastic changes following brain injury. However, the effects of such interventions to tentatively prevent secondary neurological damages have never been assessed in infants with CP. This study aims to determine the effect of the baby Hand and Arm Bimanual Intensive Therapy Including Lower Extremities (baby HABIT-ILE) in infants with unilateral CP, compared with a control intervention.

**Methods and analysis** This randomised controlled trial will include 48 infants with unilateral CP aged (corrected if preterm) 6–18 months at the first assessment. They will be paired by age and by aetiology of the CP, and randomised into two groups (immediate and delayed). Assessments will be performed at baseline and at 1 month, 3 months and 6 months after baseline. The immediate group will receive 50 hours of baby HABIT-ILE intervention over 2 weeks, between first and second assessment, while the delayed group will continue their usual activities. This last group will receive baby HABIT-ILE intervention after the 3-month assessment. Primary outcome will be the Mini-Assisting Hand Assessment. Secondary outcomes will include behavioural assessments for gross and fine motricity, visual–cognitive–language abilities as well as MRI and kinematics measures. Moreover, parents will determine and score child-relevant goals and fill out questionnaires of participation, daily activities and mobility.

**Ethics and dissemination** Full ethical approval has been obtained by the *Comité d'éthique Hospitalo-Facultaire/Université catholique de Louvain*, Brussels (2013/01MAR/069 B403201316810g). The recommendations of the ethical board and the Belgian law of 7 May 2004 concerning human experiments will be followed. Parents will sign a written informed consent ahead of participation. Findings will be published in peer-reviewed journals and conference presentations.

**Trial registration number** NCT04698395. Registered on the International Clinical Trials Registry Platform (ICTRP) on 2 December 2020 and NIH Clinical Trials Registry on 6 January 2021. URL of trial registry record: https://clinicaltrials.gov/ct2/show/NCT04698395?term=bleyenheuft&draw=1&rank=7.

## STRENGTHS AND LIMITATIONS OF THIS STUDY

⇒ This study is a randomised controlled trial (RCT) assessing motor and non-motor changes after an intensive intervention of motor activities in infants with unilateral cerebral palsy (CP) aged 6–18 months (corrected if preterm).

⇒ This study will be the first RCT to apply a bimanual and locomotor intervention in this age group and to assess neuroplastic changes after an intervention in infants with unilateral CP.

⇒ The primary outcome (Mini-Assisting Hand Assessment) and the secondary motor outcome (Gross Motor Function Measure-66) will be videotaped and blindly scored.

⇒ Recruiting 24 pairs of infants in a narrow age range who fit our inclusion criteria such as an early diagnosis of unilateral CP might be a challenge.

## INTRODUCTION

Cerebral palsy (CP), the most prevalent paediatric motor disability, is a group of disorders of movement and posture due to non-progressive brain damage in the developing brain occurring in 1 out of 500 neonates.[1] Depending on the location of the brain damage, children can present motor impairments,[2] as well as associated symptoms including sensory, cognitive and visuospatial impairments.[1] These impairments generally lead to highly variable limitations both in gross motor function and manual abilities, which may greatly impair the future execution of many activities of daily living (such as dressing, eating, toileting, etc), affecting both the lifelong autonomy and the quality of life of these children.[3–5]

As the brain damage is non-evolutive, the main focus of management of children with CP is currently the treatment of the symptoms, most of the time through neurorehabilitation based on neurodevelopmental

approaches, usually a few hours a week including a mix of stretching and movements manually guided towards 'normality'.[6] However, the efficiency of these approaches is debated.[6 7] In contrast, there is strong evidence that more recent intensive interventions based on motor skill learning principles are more efficient in inducing functional and neuroplastic changes, across age groups and disability levels, compared with traditional non-intensive neurorehabilitation.[7–13]

The intensive interventions are organised as blocks of training using motor learning concepts (practice specificity, context of learning, feedback, etc),[14] aiming to elicit practice-induced brain changes (neuroplasticity) arising from repetition, increasing movement complexity, motivation and reward.[15 16] Though efficient to improve motor abilities, these interventions remain limited to the mere treatment of symptoms.

Research using animal models[17 18] suggests that intensive motor skill training in infants with CP may significantly reduce or even prevent maladaptive neuroplastic changes consecutive to the brain lesion, potentially addressing some of the root causes of CP. More specifically, the initial brain damage induces a cascade of secondary neuroplastic damages that account for the motor disability: chronic inflammation,[19–21] alterations in the organisation of the corticospinal tract (CST)[22 23] and underdeveloped CST projections at the spinal level.[24 25] Moreover, due to the perception-action loop, the resulting motor deficit may in turn affect the developmental course of other functions such as attentional, visual and spatial cognition.[26]

Only a few motor-intensive intervention trials have been carried out in young children under 3 years old.[27] These studies have focused on children with unilateral CP and evaluated modified forms of Constraint-Induced Movement Therapy (CIMT)[28–32] and environmental enrichment intervention.[33] These studies have provided moderate support in favour of CIMT and enrichment intervention. Higher daily intensity of training[30 31] thereby seems to induce larger effects than lower daily intensity training.[28 29 32] Even though lower extremities are commonly affected in children with CP, CIMT focuses solely on the upper extremities.[28–32] Moreover, none of these studies have investigated the possible impact of these interventions on organisation and projections of the CST, as well as on non-motor functions.

Hand and Arm Bimanual Intensive Therapy Including Lower Extremities (HABIT-ILE), developed by Bleyenheuft and Gordon (2014), is an intensive intervention based on motor skill learning principles which focuses on a bimanual use associated to postural and/or gross motor stimulation.[10] Its efficiency was demonstrated for both upper and lower extremities in children with unilateral[8 34] and bilateral CP[9] from 1 to 18 years old. An impact has been observed on motor function, visuospatial abilities and improvement of brain structure in children.[35] Such an intervention has never been attempted in infants aged between 6 and 18 months with CP to tentatively prevent secondary neurological damages or developmental trajectories.

## Aims and hypotheses

This study aims to demonstrate the effectiveness and feasibility of an intensive motor skill learning intervention (baby HABIT-ILE) in preventing secondary damage and in inducing changes in motor and non-motor functions in children with unilateral CP, compared with a control intervention (usual activities). We hypothesise that the baby HABIT-ILE intervention will induce greater improvements in the immediate group compared with usual activities of the delayed group. Moreover, possible differences related to therapy onset will be observed, with a potential disadvantage for the delayed group due to the influence of a 3 months lag in receiving intensive intervention during their specific window of opportunity. A randomised controlled trial (RCT) was designed to verify our hypothesis after 2 weeks of treatment or control intervention, and at 3 months follow-up. The results, scored by experts for the tests and by parents for the questionnaires, will be correlated with MRI changes.

## METHODS AND ANALYSIS
### Study design

This RCT will follow a two-parallel group design, involving 48 infants with unilateral CP between 6 and 18 months (corrected age). Assessments will be performed at the Institute of Neuroscience of the Université catholique de Louvain (UCLouvain) in Brussels, Belgium, at three time points: baseline (T0), 1 month after baseline (T1) and 3 months after baseline (T2). The between-group comparisons will be carried out between T0 and T1 and between T0 and T2. The delayed group will receive after the third assessment the baby HABIT-ILE intervention. A supplementary assessment 6 months after baseline (T3) will allow investigating the influence of the onset of therapy between both groups, and the persistence of the therapeutic effects at long term in the immediate group. At each assessment time, participants will follow motor and non-motor, functionals skills, participation, kinematics and neuroimaging assessments (see figure 1).

The Consolidated Standards of Reporting Trials statement for non-pharmaceutical and pragmatic trials will be used for the structure of this study and the Standard Protocol Items: Recommendations for Interventional Trials guidelines were used for the writing of this protocol. The study will be conducted from February 2021 to December 2023.

### Recruitment, inclusion and exclusion criteria

Forty-eight infants will be included. They will be recruited from rehabilitation centers of Belgium. Spontaneous applications from parents will also be considered.
To be eligible for inclusion, infants must be:
► Aged (corrected age if preterm birth) 6–18 months at the first assessment.

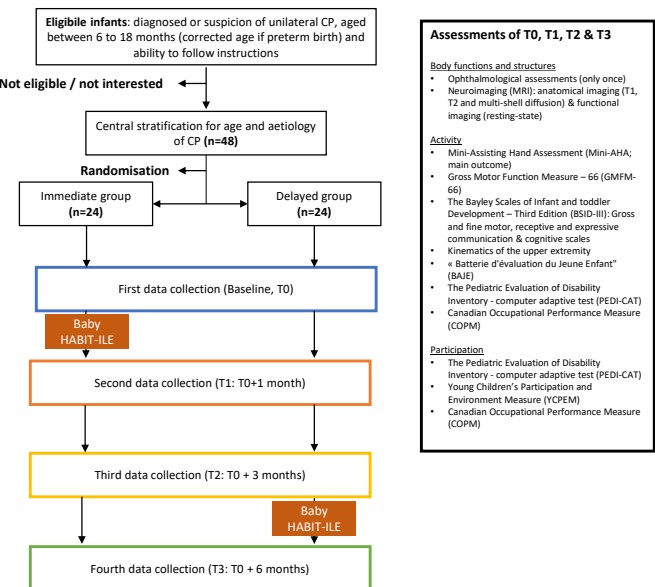

**Figure 1** Consolidated Standards of Reporting Trials Flow chart. CP, cerebral palsy; HABIT-ILE, Hand and Arm Bimanual Intensive Therapy Including Lower Extremities.

▶ Diagnosed with unilateral CP or suspicion because of asymmetry or at risk of developing unilateral CP.

▶ Able to follow instructions and complete testing according to the age.

Infants will be excluded if they have:

▶ Uncontrolled seizures.

▶ Botulinum toxin injections, orthopaedic surgery or specific intensive therapy within 6 months before and until the end of the study.

▶ Implants or other metallic devices that are contraindications to MRI.

▶ Severe visual or cognitive impairments preventing the child from being interested in simple games.

### Sample size calculation

The sample size calculation was performed based on the Assisting Hand Assessment (AHA[36]) score of a HABIT-ILE pilot study performed in 10 infants with unilateral CP.[34] In this pilot study, toddlers with unilateral CP (1–4 years old) encountered an intensive intervention of 50 hours over 2 weeks, as in the present study. A mean improvement of 8.2 AHA units (SD=5.9 AHA units) after the intensive therapy was reported. Thus, our hypothesis is an incremental improvement of 1 SD between the immediate group and the delayed group at second and third assessments (between group difference=7.2, SD=5.9). With $\alpha$=0.05 and a 1–$\beta$=0.9, a sample size of 15 participants per group is required. Considering potential dropouts and data loss in MRI, 24 participants will be included in each group (48 in total).

### Randomisation process

A pair-matched randomisation will be used. Group allocation to either immediate or delayed condition will be randomly determined within each pair using an online computer-based randomisation sequence.[37] Children will be matched regarding age (± 4 months) and aetiology of CP (brain malformation/periventricular white matter lesion/grey matter lesion; following Krägeloh-Mann's classification[2]) before the baseline assessment and collection of informed consent. If possible, children will also be matched considering the affected hemisphere (right/left).

### Blinding procedure

The participants and interventionists (therapists) will be blind to the group allocation. Study allocation codes will be used for analyses and all data will be anonymised. The Mini-AHA and the Gross Motor Function Measure-66 (GMFM-66) will be videotaped and blindly scored by external accredited/experienced raters unaware of group allocation or assessment time. Anonymised data will be stored in the Research Electronic Data Capture (REDCap) register which is hosted at the UCLouvain. Once the data collection is over, anonymised data will be analysed.

### Study interventions

#### Control intervention: usual activities

Usual activities have been chosen as control intervention since spontaneous activity of infants is oriented towards the discovery of their environment with motor activities in a nursery reaching around 5 hours per day.[38] The infants allocated to the delayed group will continue,

between the first and the second assessment, their usual activities including nursery and ongoing therapy, generally a few hours of neurodevelopmental therapy per week. Wrist movement sensors will record the amount of motor activity during 5 days, 5 hours a day, to document the percentage of total time spent in movement during the control intervention.

### Treatment intervention: baby HABIT-ILE

The intensive motor skill training consists of an adaptation of HABIT-ILE.[8–10] This intervention, which was developed initially for children with CP over 6 years old, is carried out in a camp setting, with structured tasks of increasing motor difficulty and functional activities that require the use of both hands while sustaining postural activity or activities of the lower extremities. The therapy, using games and functional activities, is delivered in a child-friendly environment. The tasks are chosen according to individualised functional goals previously defined by the parents (eg, drinking or sitting autonomously, turning the pages of a book, moving around). Although the basic principles of HABIT-ILE will be followed,[10] the toys, tasks and activities will be adapted to the age of the infants. The duration of activities as well as lower extremities/postural requests will also be adapted to each infant depending on his/her functional goals and individual characteristics (motor and attentional level, for instance). Twelve infants will undergo rehabilitation at the same time during one intensive camp of 2 weeks (four camps in total); each infant will have a full-time therapist dedicated to his/her treatment. The monitoring of the guidelines and the fidelity to the treatment principles will be ensured throughout the whole intervention by experienced HABIT-ILE supervisors. As 50–60 hours of intensive therapy are needed to induce long-term motor changes,[39 40] this project, in agreement with the usual motor activity time[38] and the need for infants to rest, will include 5 hours per day of therapy with the same therapist for a total of 50 hours (5 days/week). HABIT-ILE for infants has been tested in a pilot study which has confirmed previous evidence that this therapy intensity is both practically feasible and efficient for children at that age.[34] Infants will follow the intervention during 3 hours in the morning (with a short break for those who need a morning nap); they will subsequently have 2 hours of nap-break around noon, and will complete two additional hours of intervention in the afternoon.

### Data collection and data management

An 'assessment log sheet' including patient code, date of assessment, test procedure, mood of the child and context during the data acquisition will be used at each assessment. This information will be collected on REDCap[41 42] register. All children (immediate and delayed groups) will be seen at the four assessments times. At each assessment time, children will complete all the assessments listed below, except the ophthalmological assessment which is only carried out once.

Each assessment time will be performed during 1 day. To avoid tiredness and ensure quality testing, two blocks of maximum 1 hour 30 min of functional assessment will be spread over the day. Rest periods will be planned between blocks and parents will be present during the whole day.

### Outcomes

Outcomes of the study were chosen to cover the three levels of the International Classification of Functioning, Disability and Health: Children & Youth version (ICF-CY).[43] A daily logbook filled by the interventionists during the whole therapy time will be used to examine the feasibility of the intervention. Interventionists will write the time and the type of activities for each participant during the whole camp duration.

### Primary outcome

The Mini-AHA, as primary outcome, measures the use of the more affected hand collaboratively with the less affected hand in bimanual activities. This videotaped test targets the activity domain of the ICF-CY and uses Rasch model analysis. The score is expressed on a logit-based AHA-unit scale ranging from 0 to 100 (the higher the score, the better the ability). This assessment is responsive, reliable and valid for use in children with unilateral CP.[44]

### Secondary outcomes

These outcomes concern the three levels of the ICF-CY. First, the body functions and structures will be studied through ophthalmological and neuroimaging assessments. Second, the activity level of the children will be assessed with the GMFM-66,[45] the Bayley Scales of Infant and toddler Development-Third edition (BSID-III),[46] the kinematics assessment (Vicon 3D motion system[47]), the kinetics assessment (Xsens-dot movement sensors[48]) and the visuospatial battery "Batterie d'évaluation du Jeune Enfant" (BAJE).[49] Moreover, questionnaires including the Pediatric Evaluation of Disability Inventory—Computer Adaptive Test (PEDI-CAT)[50 51] and the Canadian Occupational Performance Measure (COPM)[52] will assess both activity and participation levels. Third, the Young Children's Participation and Environment Measure (YC-PEM),[53] for its part, will only assess the participation level.

#### *Body functions and structures assessments*

To determine neuroplastic changes, MRI datasets will be acquired. Those assessments will take place in the Saint-Luc University Hospital in Brussels using a 3 T MRI (SIGNA Premier, General Electric), equipped with a 48-channel head coil. The MRI procedure will consist in night scanning during natural, non-sedated sleep. Infants will undergo a standardised, gradual, adaptation home programme to accustom them to the specific conditions of the MRI environment (wearing earplugs and careful examination of each infant's sleep routines[54]). Adapted transfer material allowing the infant to fall asleep in a child-friendly environment in the MRI service and to

be transferred asleep in the machine will be used. The total time of MRI is estimated to 35 min. MRI changes will be measured with anatomical imaging (T1, T2 and multishell diffusion sequences) and functional imaging (resting-state sequence). More precisely, our analyses will include the classical diffusion tensor imaging indicators of fractional anisotropy, radial diffusivity, axial diffusivity and mean diffusivity[55 56] as well as cortical thickness. In addition, new advanced model-based microstructural features based on angular weighting will be used to investigate microscopic level of training-induced changes.[57 58] At macroscopic level, tractography will be used to highlight potential changes in CST and optic radiation organisation.[59 60]

All infants will encounter an ophthalmological assessment at the last assessment (T3) by ophthalmologists from the Saint-Luc University Hospital, Brussels. A standard battery of ophthalmological tests for infants will be used (Hirshberg,[61] Gracis biprism,[62] Lang II,[63 64] Cover Test,[65] ocular motility, Cardiff test,[66] empty refraction, refraction under cycloplegia, dilated fundus, etc). This assessment will allow to exclude visual impairments of peripheral origin, to document visual impairments and to describe the baseline of the ophthalmological condition of these infants.

### Activity assessments

Gross motor function will be measured using the GMFM-66.[45] This tool is a clinical assessment for children with CP allowing the measure of changes in gross motor function over time or following an intervention. The items cover a large spectrum of activities divided into five domains: 'lying and rolling', 'sitting', 'crawling and kneeling', 'standing' and 'walking, running and jumping'. This test will be videotaped and uses Rasch model analysis. The score is expressed on percentage of logits. This test is responsive,[67] reliable and valid for infants with CP.[68]

The BSID-III[46] will be used to assess cognitive, motor and language development. The cognitive scale includes attention and habituation tasks, problem-solving, play tasks, object assembly, concept grouping and memory tasks. The motor scale, divided into two subscales, includes assessments of gross and fine motor skills such as grasping, sitting, stacking blocks and climbing stairs. Finally, the language scale, also divided into two subscales (receptive and expressive communication), includes understanding and expression of language (recognition of objects and people, following directions, naming and recognising objects and pictures, etc). Raw scores of the items are converted to scaled scores and composite scores (mean 100, SD 15) ranging from 40 to 160 (higher scores indicate better performance). This assessment has been shown to be reliable and valid in young children with CP.[69]

Kinematics assessments will be used to determine qualitative changes in upper extremities movement characteristics. Markers will be positioned on the head (using a headband), acromions, elbows and wrists of the child. Children will sit on an adapted chair, in front of a table. If the child does not sit up independently, an adult will support his/her trunk. The first task will consist of a unimanual reaching of a toy, with the less-affected hand and the more-affected hand, separately. The second task will be bimanual: children will be asked to tap two cymbals against each other. Through a 3D motion system (Vicon Motion Systems, Oxford, UK[47]), the straightness (percentage of upper extremities trajectory during a reaching task) and smoothness (variability of movement during a reaching task) of the upper extremities will be studied for each task. Also, the time of activity (time from onset to end of the task in seconds) during a reaching task will be analysed.

Quantitative changes of physical activity will also be measured with two movement sensors (Xsens-Dot[48]), placed on each wrist during 5 hours a day for 5 days at home and during the camp duration (documented by a logbook). The percentage of total time spent in movement (ie, crawling, walking and running) will be measured and calculated in terms of the changes in the acceleration ($m/s^2$).

In addition, the visuospatial battery BAJE[49] will be used to assess changes in visuospatial attention and visuoattentional abilities.[70] The BAJE is an adaptation of the evaluation of visuospatial abilities[70–72] which is designed to test visuospatial and visuoattentional abilities of children from 3 months of age, developed and validated by Sylvie Chokron's team. This normalised and standardised battery includes simple tasks such as photomotor reflex and blink reflex at threat, visual fixation, light detection, visual pursuit, visual filed/close visual field and visuomotor coordination, and so on. In this study, through three different tasks of this battery (visuomotor coordination, visual field and eye pursuit), we will evaluate the presence and severity of the most common cerebral visual impairments (visual field defect, visuoattentional, visuospatial and visuomotor deficits).

The 'daily activities' and 'mobility' domains of the PEDI-CAT will be used to evaluate the capacity of upper and lower extremities of children with CP during specific activities.[50] The scores are computed through the PEDI-CAT software (scores reported in scaled scores). This questionnaire is sensitive for children with CP.[73]

The COPM,[52] a semistructured interview performed by a trained examiner, will be used to define the therapeutic goals and to quantify the child's performance (1–10 scale) in relation to these goals, as well as the satisfaction (1–10 scale) of the parents regarding their child's achievement. Total score is the average of the scores for performance and satisfaction separately.

### Participation assessments

Participation will be evaluated using the PEDI-CAT,[74] the COPM[52] and the YC-PEM.[53] Based on different children's activities, the YC-PEM is a parents-filled questionnaire

evaluating the level of participation and the quality of the environment across three settings: home, daycare/preschool and community. For each type of activity, caregivers assess multiple dimensions of the child's participation: frequency, level of involvement, caregiver's percent desire for change and perceived impact of environmental support. It has a good intrarater and inter-rater reliability.[53]

## Statistical analysis

A two-way repeated measures analysis of variance (2 groups (immediate vs delayed) × 3 assessment times (T0, T1, T2)) with repeated measures on the time factor and Holm–Šidák post hoc analyses for pairwise multiple comparisons (or non-parametric equivalent when normality and homoscedasticity are not met) will be used for the primary outcome, the Mini-AHA, and the secondary outcomes. Significant differences will be statistically set at $\alpha=0.05$.

## Ethics and dissemination

Full ethical approval has been obtained for this study protocol by the Comité d'éthique Hospitalo-Facultaire/UCLouvain, Brussels (reference number: 2013/01MAR/069 B403201316810g). An information letter will be provided to the parents. They will be asked to sign a consent form if they comply to their child's participation. The parents will be informed during the recruitment process of their right to withdraw their child at any time without prejudice. All data collected will be anonymous and will be stored by the UCLouvain. Findings from this study will be published in peer-reviewed scientific journals and conference presentations. This study has been registered on NIH Clinical Trials Registry on 6 January 2021 (NCT04698395, https://clinicaltrials.gov/ct2/show/NCT04698395?term=bleyenheuft&draw=1&rank=7). After contacting the corresponding author and signing a data sharing agreement, a digital object identifier could be shared to access selected data.

## Patient and public involvement

Patients and the public were not involved in the design, or conduct, or reporting, or dissemination plans of this research.

## CONCLUSION

This RCT has been designed to substantiate evidence of the efficacy of the baby HABIT-ILE intervention to provide functional and neuroplastic changes in infants with unilateral CP. By implementing the baby HABIT-ILE intervention very early in the child's development during a specific window of opportunity, we hope to prevent the onset of secondary damages and, consequently, to potentiate the long-term motor abilities of the children and increase their autonomy.

**Author affiliations**
[1]Institute of Neuroscience, Université catholique de Louvain, Brussels, Belgium
[2]Neurology Department, Stroke Unit/Motor Learning Lab, CHU UCL Namur, Yvoir, Belgium
[3]Exercise and Rehabilitation Science Institute, School of Physical Therapy, Faculty of Rehabilitation Science, Universidad Andrés Bello, Santiago, Chile
[4]Louvain Bionics, Université catholique de Louvain, Louvain-la-Neuve, Belgium
[5]Institute of Information and Communication Technologies, Electronics and Applied Mathematics (ICTM), Université catholique de Louvain, Louvain-la-Neuve, Belgium

**Contributors** YB, LD, BM and YV contributed to study design and obtained funding for the research study. ACdT and EH first drafted this protocol. YB, RA, DE-K, EH, ACdT and EG will be involved in the implementation of the therapy. EH, ACdT, EG, DE-K and RA will recruit the participants and perform the assessments. YB, ACdT, EH, LD, DE-K and RA will contribute to data analysis. All authors contributed to the writing of this manuscript and have critically reviewed and approved the final version.

**Funding** This study is part of a larger study funded by the Fédération Wallonie-Bruxelles through the program "Action de Recherche Concertée" of the UCLouvain (20/25-105), Belgium. Partial grant specifically dedicated to the visual part has also been provided by the "Fondation JED-Belgique". EH has a grant from the "Fonds de la Recherche Scientifique - FNRS" (A1.21302.026-F). The funding bodies have reviewed this protocol during the selection process, but after this initial point, have participated neither in the study design nor in data collection, analysis, interpretation data, nor in writing the manuscript.

**Competing interests** None declared.

**Patient and public involvement** Patients and/or the public were not involved in the design, or conduct, or reporting, or dissemination plans of this research.

**Patient consent for publication** Not applicable.

**Provenance and peer review** Not commissioned; externally peer reviewed.

**ORCID iDs**
Astrid Carton de Tournai http://orcid.org/0009-0002-2409-8711
Enimie Herman http://orcid.org/0009-0005-1042-7985
Estelle Gathy http://orcid.org/0000-0002-9878-2420
Daniela Ebner-Karestinos http://orcid.org/0000-0002-5941-8706
Rodrigo Araneda http://orcid.org/0000-0002-0890-5504
Laurence Dricot http://orcid.org/0000-0002-8613-2420
Benoît Macq http://orcid.org/0000-0002-7243-4778
Yannick Bleyenheuft http://orcid.org/0000-0002-6688-932X

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
