## [Reviewer comments · BMJ Open]

ARTICLE DETAILS

TITLE (PROVISIONAL)	Baby HABIT-ILE intervention: study protocol of a randomized controlled trial in infants aged 6-18 months with unilateral cerebral palsy
AUTHORS	Carton de Tournai, Astrid; Herman, Enimie; Gathy, Estelle; Ebner-Karestinos, Daniela; Araneda, Rodrigo; Dricot, Laurence; Macq, Benoît; Vandermeeren, Yves; Bleyenheuft, Yannick

VERSION 1 – REVIEW

REVIEWER	Pinero-Pinto, Elena Universidad de Sevilla, Departamento de Fisioterapia
REVIEW RETURNED	11-Oct-2023

GENERAL COMMENTS	Thank you for the opportunity to review this manuscript on a HABIT-ILE protocol in a very young population. It is the first, so far, of these characteristics. Although the study is already being carried out and is expected to be completed in December 2023, we eagerly await its results. There are no suggestions for improvement, since the methodology is well described, very complete in terms of the variables to be analyzed and a good implementation proposal. Congratulations.
--

REVIEWER	Janssen-Potten, Yvonne Maastricht University, Department of Rehabilitation Medicine, School for Public Health and Primary Care (CAPHRI)
REVIEW RETURNED	06-Nov-2023

GENERAL COMMENTS	This is a thorough study protocol of a very relevant study that is nearing completion. I read it with great pleasure and interest and only have some minor questions for clarification. These I have articulated in the attached document. I wish the authors every success in completing the trial and look forward to the results The following sections of the protocol article would, in my view, benefit from more clarification: Aim & hypotheses page 6: “A randomized controlled trial (RCT) was designed to verify our hypothesis immediately after 2 weeks of therapy and at 3 months follow up.” I wonder whether the phrasing “immediately after” is correct here, because measurements are scheduled at 1 and 3 months after the baseline measurement. In other words T1 is 2 weeks after the end of the HABIT-ILE intervention for half of the immediate group. The
---

	immediate effect in the delayed group is even more difficult to measure as T3 could be up to 10 weeks after the end of the intervention for the children of the 3rd camp. Page 6 studie design: Assessment will be performed..... at three intervals....” The term intervals may lead the reader astray. Because subsequently only three measuring moments are mentioned and two intervals will be tested, namely T0-T1 and T0-T2. The fourth measurement moment is only introduced later and this then probably also provides the third interval but which interval this is, is not explicitly named. By long-term effects, do the authors mean T0-T3? Or are they testing the extent to which any improvements persist T1-T3? A lot of measurements are taken, this seems quite stressful for the very young participants. Can the authors explain a little more about how this is handled? Is testing done in blocks? If so, how much time/days in between? Blinding procedure, page 8: “The participants and interventionists (therapists) will be blind to the group allocation.” The results of on the COPM and the CY-PEM are based on parents' judgment. Parents are not blinded, they know when their child participates in the therapy camp after the baseline measurement and thus whether their child is in the immediate or delayed group. Treatment goals are set at baseline using the COPM, the disadvantage of this is that parents of children in the delayed group should not focus on these goals for three months in their home situation. Outcome measurements, page 13: “During the kinematic assessments children will sit on an adapted chair, in front of a table.” How is this handled if the child cannot yet sit up (independently)? “Physical activity will be measured with two movement sensors, placed on each wrist during 5 hours a day for 5 days at home and during the camp duration.” I can imagine that therapists are more focused on putting the sensors on at the start of the day's programme (3 hours in the morning and 2 hours in the afternoon) than the parents. Especially when you think of working parents, a child going to a childcare facility or grandparents looking after the child during the day. Is this protocolised? Statistical analysis, page 15: The static analysis is described very briefly. How many covariables are included in the analysis? If more intervals are tested, why is no correction applied? And how will the differences in timing of measurements for participants in the 4 therapy camps be dealt with (see also immediate effects)?
--	--

VERSION 1 – AUTHOR RESPONSE

Reviewer 1: Dr. Elena Pinero-Pinto, Universidad de Sevilla

Thank you for the opportunity to review this manuscript on a HABIT-ILE protocol in a very young population. It is the first, so far, of these characteristics.

Although the study is already being carried out and is expected to be completed in December 2023, we eagerly await its results.

There are no suggestions for improvement, since the methodology is well described, very complete in terms of the variables to be analyzed and a good implementation proposal. Congratulations.

- Authors: Thank you for your encouraging comment.

Reviewer 2: Dr. Yvonne Janssen-Potten, Maastricht University, Adelante

This is a thorough study protocol of a very relevant study that is nearing completion. I read it with great pleasure and interest and only have some minor questions for clarification. These I have articulated in the attached document. I wish the authors every success in completing the trial and look forward to the results.

- Authors: Thank you for your relevant comments and positive feedback.

The following sections of the protocol article would, in my view, benefit from more clarification:

Aim & hypotheses page 6: “A randomized controlled trial (RCT) was designed to verify our hypothesis immediately after 2 weeks of therapy and at 3 months follow up.”

I wonder whether the phrasing “immediately after” is correct here, because measurements are scheduled at 1 and 3 months after the baseline measurement. In other words T1 is 2 weeks after the end of the HABIT-ILE intervention for half of the immediate group. The immediate effect in the delayed group is even more difficult to measure as T3 could be up to 10 weeks after the end of the intervention for the children of the 3rd camp.

- Authors: Thank you, we agree and we have clarified (first paragraph page 6: [A randomized controlled trial (RCT) was designed to verify our hypothesis after 2 weeks of treatment or control intervention, and at 3 months follow up].

Page 6 studie design: Assessment will be performed..... at three intervals....”

The term intervals may lead the reader astray. Because subsequently only three measuring moments are mentioned and two intervals will be tested, namely T0-T1 and T0-T2.

- Authors: Thank you for your comment. We corrected it with the term “time points” instead of “intervals” (last paragraph page 6: [Assessments will be performed at the Institute of Neuroscience of the Université catholique de Louvain (UCLouvain) in Brussels, Belgium, at 3 time points: baseline (T0), 1 month after baseline (T1) and 3 months after baseline (T2)].

The fourth measurement moment is only introduced later and this then probably also provides the third interval but which interval this is, is not explicitly named. By long-term effects, do the authors mean T0-T3? Or are they testing the extent to which any improvements persist T1-T3?

- Authors: Thank for the comment. We modified in the text in order to better explain. T3 will be used to compare the influence of the onset of therapy between both groups (immediate and delayed). Moreover, T3 will allow us to investigate the persistent effects at long term for the immediate group (last paragraph page 6: [A supplementary assessment 6 months after baseline (T3) will allow investigating the influence of the onset of therapy between both groups, and the persistence of the therapeutic effects at long term in the immediate group]).

A lot of measurements are taken, this seems quite stressful for the very young participants. Can the authors explain a little more about how this is handled? Is testing done in blocks? If so, how much time/days in between?

- Authors: Thank you for your relevant comment. We agree that the assessments can be demanding and stressful for infants. We have added details regarding the planning of the assessments in the text (third paragraph page 10: [Each assessment time will be performed during one day. To avoid tiredness and ensure quality testing, 2 blocks of maximum 1h30 of functional assessment will be spread over the day. Rest periods will be planned between blocks and parents will be present during the whole day]).

Blinding procedure, page 8: "The participants and interventionists (therapists) will be blind to the group allocation."

The results of on the COPM and the CY-PEM are based on parents' judgment. Parents are not blinded, they know when their child participates in the therapy camp after the baseline measurement and thus whether their child is in the immediate or delayed group. Treatment goals are set at baseline using the COPM, the disadvantage of this is that parents of children in the delayed group should not focus on these goals for three months in their home situation.

- Authors: Thank you for the comment. It is true parents do know when their children will get the therapy but with the perception for the delayed group that we want to document their natural evolution before intervention. And yes during that time the children can evolve naturally or likely due to parental / regular therapy in their goals. That is actually the interest of having a group without intensive intervention, helping to document the spontaneous evolution.

Outcome measurements, page 13:

"During the kinematic assessments children will sit on an adapted chair, in front of a table."

How is this handled if the child cannot yet sit up (independently)?

- Authors: We clarified in the text (third paragraph page 13: [If the child does not sit up independently, an adult will support his/ her trunk]).

"Physical activity will be measured with two movement sensors, placed on each wrist during 5 hours a day for 5 days at home and during the camp duration."

I can imagine that therapists are more focused on putting the sensors on at the start of the day's programme (3 hours in the morning and 2 hours in the afternoon) than the parents. Especially when you think of working parents, a child going to a childcare facility or grandparents looking after the child during the day. Is this protocolised?

- Authors: Thank you for your comment. To try to counter this, parents will be asked to fill in a logbook, mentioning the time the sensors will be put on and take off, and the activities carried out while the child wears the sensors (first line page 14: [Quantitative changes of physical activity will also be measured with two movement sensors (Xsens-Dot (48)), placed on each wrist during 5 hours a day for 5 days at home and during the camp duration (documented by a logbook)]).

Statistical analysis, page 15:

The static analysis is described very briefly. How many covariables are included in the analysis? If more intervals are tested, why is no correction applied? And how will the differences in timing of measurements for participants in the 4 therapy camps be dealt with (see also immediate effects)?

- Authors: After discussions with statisticians, it turns out that repeated measures analysis of variance tests would be more appropriate because infants will be paired at recruitment considering several factors, so no covariables will be needed (second paragraph page 15: [A two-way repeated measures analysis of variance (RM ANOVA) (2 groups [immediate vs delayed] x 3 assessment times [T0, T1, T2]) with repeated measures on the time factor and Holm– Šidák post hoc analyses for pairwise multiple comparisons (or non-parametric equivalent when normality and homoscedasticity are not met) will be used for the primary outcome, the Mini-AHA, and the secondary outcomes]).